# Achieving net zero greenhouse gas emissions critical to limit climate tipping risks

Tessa Möller [1,2,3,4,5,12] ✉, Annika Ernest Högner [3,4,5,12],
Carl-Friedrich Schleussner [1,2,6], Samuel Bien [3,4,5], Niklas H. Kitzmann [3,4],
Robin D. Lamboll [7], Joeri Rogelj [1,7,8], Jonathan F. Donges [3,9,10],
Johan Rockström [3,5,9] & Nico Wunderling [3,10,11] ✉

Under current emission trajectories, temporarily overshooting the Paris global warming limit of 1.5 °C is a distinct possibility. Permanently exceeding this limit would substantially increase the probability of triggering climate tipping elements. Here, we investigate the tipping risks associated with several policy-relevant future emission scenarios, using a stylised Earth system model of four interconnected climate tipping elements. We show that following current policies this century would commit to a 45% tipping risk by 2300 (median, 10–90% range: 23–71%), even if temperatures are brought back to below 1.5 °C. We find that tipping risk by 2300 increases with every additional 0.1 °C of overshoot above 1.5 °C and strongly accelerates for peak warming above 2.0 °C. Achieving and maintaining at least net zero greenhouse gas emissions by 2100 is paramount to minimise tipping risk in the long term. Our results underscore that stringent emission reductions in the current decade are critical for planetary stability.

Climate tipping elements are complex subsystems of the Earth system that can display non-linear, often abrupt transitions in response to anthropogenic global warming[1,2]. This means that a small increase in global mean temperature (GMT) can trigger a large qualitative change in these subsystems. Decreasing the forcing back to its pre-industrial value will often not reverse this change, as the transitions are driven by self-amplifying feedback mechanisms that lead to hysteresis behaviour[3,4].

Core tipping elements with planetary-scale impacts on the Earth system include cryosphere subsystems such as the Greenland Ice Sheet (GIS) and the West Antarctic Ice Sheet (WAIS), large-scale oceanic and atmospheric circulation patterns such as the Atlantic Meridional Overturning Circulation (AMOC), and biosphere subsystems like the Amazon Rainforest (AMAZ), the four of which we will focus on in this study. Further tipping elements include Boreal Permafrost, extra-polar mountain glaciers, and tropical coral reefs, among others[2]. Many of these tipping elements are connected through interaction processes that can stabilise or exacerbate their individual dynamics[5,6], potentially enabling tipping cascades[7]. This depends on the strength of the interactions and sensitivity to increases in GMT. Consequences of climate tipping would be severe and potentially include a global sea level rise of several metres, ecosystem collapse, widespread biodiversity loss, and substantial shifts in global heat redistribution and precipitation patterns[8]. Paleorecords, as well as

[1]Energy, Climate and Environment Program, International Institute for Applied Systems Analysis (IIASA), Laxenburg, Austria. [2]Climate Analytics, Berlin, Germany. [3]Earth System Analysis, Potsdam Institute for Climate Impact Research (PIK), Member of the Leibniz Association, Potsdam, Germany. [4]Institute of Physics and Astronomy, University of Potsdam, Potsdam, Germany. [5]Institute of Environmental Science and Geography, University of Potsdam, Potsdam, Germany. [6]Geography Department & IRI THESys, Humboldt University of Berlin, Berlin, Germany. [7]Centre for Environmental Policy, Imperial College London, London, UK. [8]Grantham Institute for Climate Change and the Environment, Imperial College London, London, UK. [9]Stockholm Resilience Centre, Stockholm University, Stockholm, Sweden. [10]High Meadows Environmental Institute, Princeton University, Princeton, NJ, USA. [11]Center for Critical Computational Studies (C³S), Goethe University Frankfurt, Frankfurt am Main, Germany. [12]These authors contributed equally: Tessa Möller, Annika Ernest Högner.
✉e-mail: moeller@iiasa.ac.at; nico.wunderling@pik-potsdam.de

observational and model-based studies, provide evidence of the multistability and hysteresis behaviour of single tipping elements[1,2]. In spite of this, most state-of-the-art high-dimensional earth system models (ESMs) do not yet comprehensively simulate the non-linear behaviour, feedback, and interactions between some of the tipping elements due to computational limitations and a lack of processes important for resolving tipping[9–11]. Most state-of-the-art ESMs do not include coupled dynamic ice sheets, which renders them unable to represent the tipping point dynamics of cryosphere tipping elements as well as their links and interactions with other tipping elements[12]. The resulting lack of freshwater forcing and sea level rise can have significant repercussions for the behaviour of ocean circulations in these models. For these reasons, these models may not be suited to fully resolve tipping dynamics and interactions[9–11].

A simplified but established complementary approach that we also utilise in this study is, therefore, to model tipping with fold-bifurcation models[6,7,13,14] (see Fig. 1). These conceptual models display hysteresis properties and tipping when a critical threshold is passed. The parameters of such conceptual models are based on process-understanding of the governing feedbacks of the tipping elements, such as Stommel's salt-advection feedback for the AMOC or the melt-elevation feedback for the GIS[3,8,10]. They can be found to produce stability landscapes for single tipping elements similar to more complex domain-specific models (see Supplementary Fig. 1).

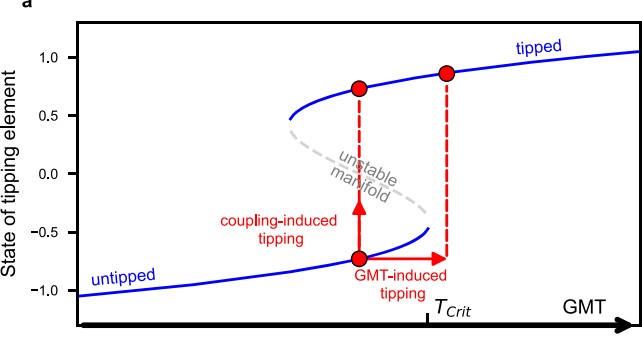

Fig. 1 | **Tipping risk and interacting tipping elements. a** Schematic fold-bifurcation diagram of a model tipping element with global mean temperature (GMT) as a forcing parameter and two stable states separated by the unstable manifold. The red arrows indicate the feedback direction of the entire system if a forcing occurs. This means, that if the system is pushed across the unstable manifold, it will move towards the opposite stable equilibrium state. **b** Illustrative time-evolution of one sample model run of each tipping element: Greenland Ice Sheet (GIS), West Antarctic Ice Sheet (WAIS), Atlantic Meridional Overturning Circulation (AMOC), Amazon Rainforest (AMAZ), including the threshold for state evaluation (dashed grey line).

It has been argued that the tipping behaviour of the GIS is linked to an ice sheet volume threshold[15,16]. However, it has also been shown that this volume threshold can be linked to a GMT threshold[17]. Similarly, the tipping behaviour of the AMOC may be primarily linked to the rate of freshwater input into the North Atlantic and AMAZ tipping behaviour has been linked to the lack of sufficient moisture supply[18]. For consistency, we have here linked the bifurcation behaviour of the tipping elements back to a GMT threshold based on multiple lines of evidence, including Earth system modelling and paleoclimate data[2].

The urgency to understand and minimise climate tipping risks has been recognised in international climate policy for the first time at the 27th Conference of the Parties (COP27) in Egypt[19]. While uncertainties are still considerable, current best estimates find several tipping elements at risk at 1.5 °C above pre-industrial GMT levels[2] and early warning signals of an approaching transition have been observed for a number of tipping elements[20–23]. This provides strong scientific support for the Paris Agreement's Article 2.1 long-term temperature goal (LTTG) aiming to limit the global temperature increase to 1.5 °C above pre-industrial levels[24], which evidence increasingly shows is a limit, not an aspirational goal[2,25]. Global warming has reached 1.2 °C[26], and current climate policy scenarios are estimated to result in 2.6 °C warming above pre-industrial levels[27] by the end of this century (with a range of 1.7–3.0 °C). Even if GMT were to be stabilised at or below 1.5 °C in the long term, a temporary overshoot above 1.5 °C is a distinct possibility and was presented prominently as the first of the Ten New Insights in Climate Science 2023/24[28], underlining the urgency that potential impacts and associated risks of such an overshoot, including the triggering of potential tipping processes, need to be assessed[29].

Previous studies have schematically analysed how individual and interacting tipping elements[7] respond to idealised overshoot scenarios[13,30], assessing the impacts of overshoot duration, peak temperature, and long-term stabilisation temperature on tipping risks. Uncertainties in critical temperatures and critical transition times, as well as−where applicable−interactions between tipping elements were incorporated. However, to systematically assess tipping risks (see Fig. 1) under a given climate policy and emission pathway, the uncertainty of the climate system in response to increasing atmospheric $CO_2$ levels (climate sensitivity and carbon-cycle feedbacks) must be taken into account[25,31]. Here, we use the PROVIDEv1.2 scenario overshoot pathways[32]−an extended version of the illustrative pathways identified in the IPCC Sixth Assessment Report[33]. The considered emission pathways span a range of different possible policies, including pathways that follow current policies and pledges, as well as pathways consistent with the climate objectives of the Paris Agreement. We study the full range of GMT outcomes for each emission pathway using multiple calibrations of the stylised Earth system model PyCascades of four interacting tipping elements[34] (GIS, WAIS, AMOC, and AMAZ), to assess tipping risks in the medium term (until 2300) and long term (in equilibrium, here after 50,000 years).

## Results

### Tipping risks under overshoots

The PROVIDEv1.2 emission pathways[32] cover the time from 1850 to 2300, harmonised to 2015 emission levels. GMT trajectories were derived using FaIR v.1.6.2[35] and extended linearly beyond 2300 to analyse long-term equilibrium behaviour. The mitigation objective, as set out in Article 4.1 of the Paris Agreement, aims to support the achievement of the LTTG by establishing a global requirement to achieve net zero greenhouse gas (NZGHG) emissions (aggregated using Global Warming Potential over a 100-year horizon, or GWP100) in the second half of the 21st century[36]. This would lead to a declining GMT[37–39]. Scenarios that achieve net zero or negative emissions by 2100 and maintain them thereafter are classified as NZGHG emission scenarios. Table 1 contains the names and properties of all analysed

**Table 1 | Scenario classification**

| Scenario | Overshoot peak temperature | NZGHG | Stabilisation temperature | Scenario assumptions |
|---|---|---|---|---|
| CurPol-OS-1.5 C | 3.30 °C | No-NZGHG | 1.5 °C | Follows current (2020) policies until 2100, then declines |
| ModAct-OS-1.5 C | 2.69 °C | No-NZGHG | 1.5 °C | Follows current (2020) pledges (NDCs) until 2100, then declines |
| ModAct-OS-1C | 2.69 °C | No-NZGHG | 1.0 °C | Follows current (2020) pledges (NDCs) until 2100, then declines |
| Ref-1p5 | – | Not defined | 1.5 °C | Reference scenario designed in temperature space |
| SSP5-3.4-OS | 2.35 °C | No-long-term-NZGHG | 1.5 °C | Tests system response to rapid emission changes |
| SSP1-1.9 | 1.53 °C | No-long-term-NZGHG | 1.0 °C | Sustainable development, no long-term compensation for non-$CO_2$ emissions |
| GS-NZGHG | 1.70 °C | NZGHG | Pre-industrial | Gradual strengthening, returns warming to 1.5 °C by 2215 |
| SP-NZGHG | 1.57 °C | NZGHGP | Pre-industrial | Broad shift towards sustainable development |
| Neg-NZGHG | 1.67 °C | NZGHG | Pre-industrial | Returns warming to 1.5 °C by 2100 with heavy CDR deployment |
| Neg-OS-0C | 1.67 °C | NZGHG | Pre-industrial | Returns warming to 1.5 °C by 2100 with heavy CDR deployment |

The 10 analysed PROVIDEv1.2 emission pathways were classified according to their median overshoot peak temperature and achievement of net zero greenhouse gas emissions (NZGHG) (NZGHG': reach NZGHG emissions by 2100 and maintain NZGHG emissions in the long term; 'No-long-term-NZGHG': reach NZGHG emissions by 2100 but do not maintain NZGHG emissions in the long term; 'No-NZGHG': do not reach NZGHG emissions by 2100), median long-term stabilisation temperatures, and scenario assumptions. Stabilisation temperature is given as median, NDCs denote Nationally Determined Contributions, and CDR denotes Carbon Dioxide Removal. For details on the NZGHG classification see Methods.

scenarios. The criteria for classification are described in the "Methods" section in more detail.

A comprehensive risk assessment requires consideration of the combined risks[25] of uncertainties on future emission trajectories, uncertainties in the Earth system response to these emissions including climate sensitivity and carbon-cycle feedbacks, as well as uncertainties regarding the tipping dynamics (see Fig. 2). Therefore, all considered scenarios take the 10–90% emission-temperature uncertainty into account, which arises from the uncertainties in the carbon cycle and climate response (see Supplementary Fig. 2). The tipping-related uncertainties are propagated via a Monte Carlo ensemble approach (see the "Methods" section).

We find that tipping risks until 2300 are substantial for several of the assessed scenarios (see Fig. 3a, b). In the long term, an overall increase in tipping risk is observed. The five pathways that do not return warming to below 1.5 °C by 2100 (CurPol-OS-1.5C, Mod-Act-OS-1.5C, Mod-Act-OS-1.0C, SSP5-3.4-OS, GS-NZGHG) display the highest risks in the medium term (Fig. 2a), reaching 23–71% tipping risk for the scenario following current (2020) policies (median 45%; CurPol-OS-1.5C). The two pathways with less than 0.1 °C median overshoot above 1.5 °C display the lowest tipping risks in the medium term with 0–7% tipping risk (median < 1%; SP-NZGHG, SSP1-1.9). If warming is returned to below 1.5 °C by 2100 after a high overshoot (median peak temperature exceeds 1.5 °C by more than 0.1 °C), tipping risks remain at or below 10% (median 2%; Neg-OS-0C and Neg-NZGHG). Failing to return warming below 1.5 °C by 2100, despite reaching NZGHG in this time, results in tipping risks of 0–24% (median 4%; GS-NZGHG). This confirms that the risks of overshoot can be minimised if warming is swiftly reversed. However, this would require rapid employment of appropriate mitigation measures.

In the long term, stabilisation temperature is one of the decisive variables for tipping risks (Fig. 2d). We find that a long-term temperature stabilisation at 1.5 °C even without prior overshoot (Ref1p5) results in more than 50% tipping risk.

Only the three scenarios that return median warming to below 1.5 °C by 2100 and maintain NZGHG thereafter (SP-NZGHG, Neg-NZGHG, Neg-OS-0C) retain long-term median risks in the very unlikely range, and upper risks below 12%.

**Fast tipping elements determine medium-term tipping risks**

In Fig. 4, we show the medium- and long-term tipping risks for each of the four considered tipping elements. In the medium term the two faster tipping elements, AMOC (tipping time: 15–300 years) and AMAZ (tipping time: 50–200 years) display the highest risks while tipping remains below 11% for the two slow-onset tipping elements, GIS (tipping time: 1000–15,000 years) and WAIS (tipping time: 500–13,000 years). In the long term, risks are highest for AMOC and WAIS. Given the threshold ranges of both ice sheets, we would expect comparable outcomes for the GIS and WAIS; however, the tipping risk for GIS is significantly lower than for WAIS: Given its lower tipping timescale, the WAIS is anticipated to tip faster than the GIS for similar temperature overshoots. Additionally, a tipping AMOC would lead to strong cooling over the GIS and potentially stabilise it (see Fig. 2c). Such strong stabilising effects are improbable to exist for the WAIS according to the newest literature[40].

As we see a comparatively little increase in tipping risk from the medium term to long term for AMAZ, we conclude that AMAZ tipping is mainly caused by the overshoot itself.

The median tipping risk for the WAIS under SSP1-1.9 increases from <1% (medium-term) to 13% (long-term), and for the upper percentile from <1% to 52%, although the temperature converges below 1.5 °C. This can be explained by the fact that the tipping threshold ranges for the ice sheets begin well below 1.5 °C[2] (see Fig. 2d).

Ref-1p5 illustrates the tipping risks if peak temperature were limited to 1.5 °C and kept constant thereafter, excluding a temporary

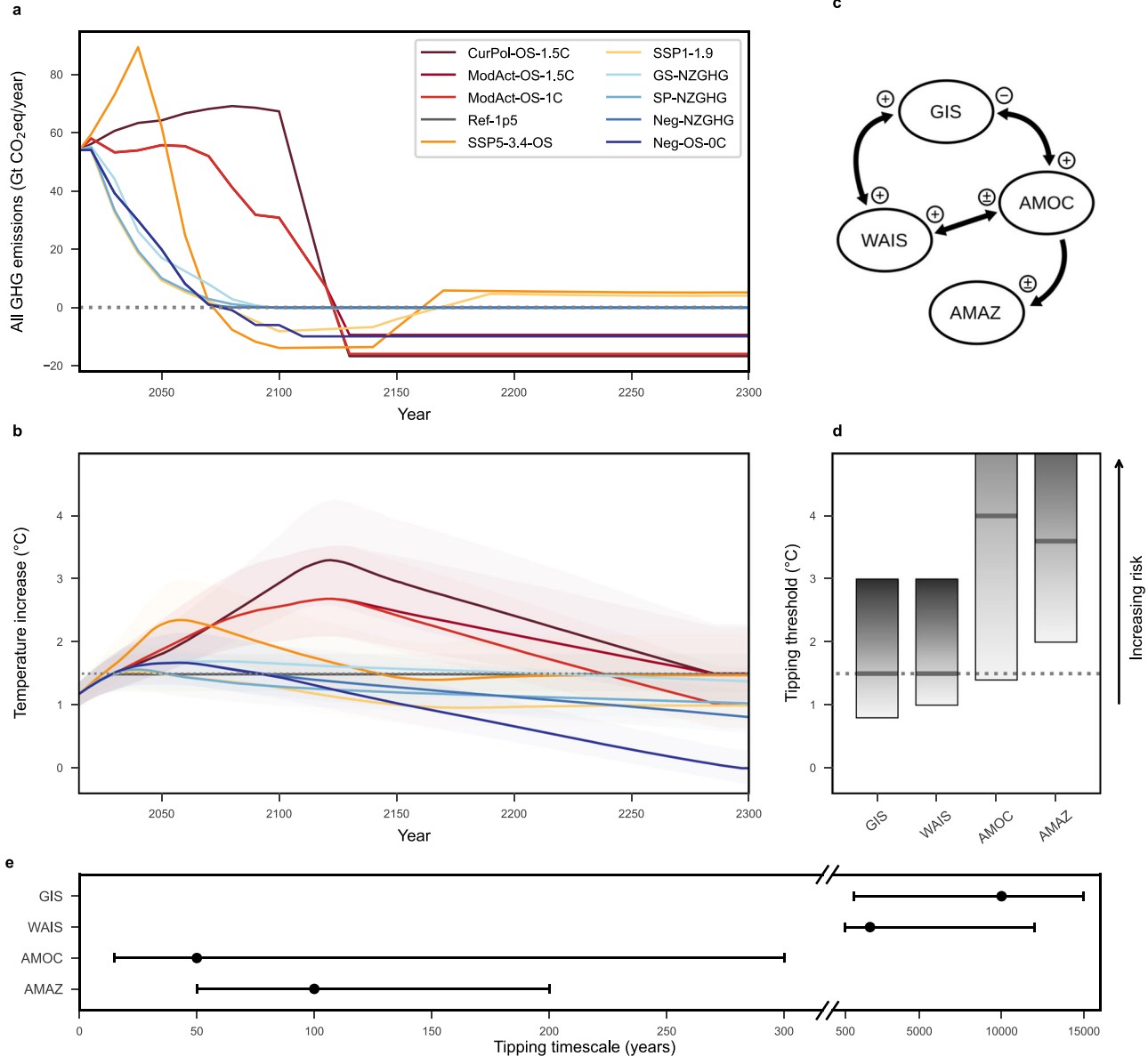

**Fig. 2 | Overview of the input data and dimensions of uncertainty. a** All-sector total greenhouse gas (GHG) emissions for nine investigated scenarios (GHG emissions as considered by the Kyoto Protocol, aggregated with Global Warming Potentials over a period of 100 years, GtCO2eq/year). **b** Resulting temperature outcomes, including climate response uncertainty, given in °C relative to pre-industrial (1850–1900 average). Shaded areas correspond to the 10–90th temperature percentiles, the median is given by the line. Scenario Ref-1p5 has been added for comparison and is only defined in temperature space. **c** Network of the four investigated tipping elements with interactions: Greenland Ice Sheet (GIS), West Antarctic Ice Sheet (WAIS), Atlantic Meridional Overturning Circulation

(AMOC), Amazon Rainforest (AMAZ). Every arrow symbolises a physical interaction mechanism between two tipping elements, categorised as destabilising (+), stabilising (−), or uncertain (±). **d** Critical temperature ranges under sustained warming for at least the respective tipping timescale, given in °C relative to preindustrial. The ranges of AMOC and AMAZ extend beyond the plot up to 8.0 and 6.0 °C, respectively. Intensifying grey indicates an increasing risk that a threshold will be exceeded, with lines marking the centre estimates. **e** Timescales of the tipping elements, with centre estimate (dot) and estimated range, from committing the tipping until it is completed. For critical temperature ranges, timescales of tipping, and interactions between tipping elements, also see Supplementary Tables 1 and 2.

overshoot as the cause for tipping. Tipping risks in the medium term under Ref-1p5 are below 10% for all elements, however they significantly increase in the long term.

### Tipping risk by 2300 from overshooting 1.5 °C
Due to different underlying mitigation assumptions, the scenarios included in this study cross the 1.5 °C limit at different times and follow different pathways to their peak and stabilisation temperatures (see Fig. 2b). To consider the impact of these pathways in more detail, we treat the temperature trajectories for each scenario as individual data points, focusing the analysis on the temperature space. We assess the

tipping risk per peak temperature for all trajectories that temporarily exceed 1.5 °C (Fig. 5a).

We find that tipping risk increases with peak warming above 1.5 °C (Fig. 5a). To further investigate this increase in tipping risk, we apply a sliding window analysis across all overshoot trajectories (Fig. 5b). Overall, the increase in tipping risk per additional 0.1 °C mean overshoot peak temperature per sliding window lies within a range of around 1.0–1.5% (Fig. 5b) for mean peak temperatures below 2.0 °C, then notably accelerates until a mean peak temperature of about 2.5 °C, above which our analysis suggests a stabilisation of the increase in tipping risk per 0.1 °C above 3%.

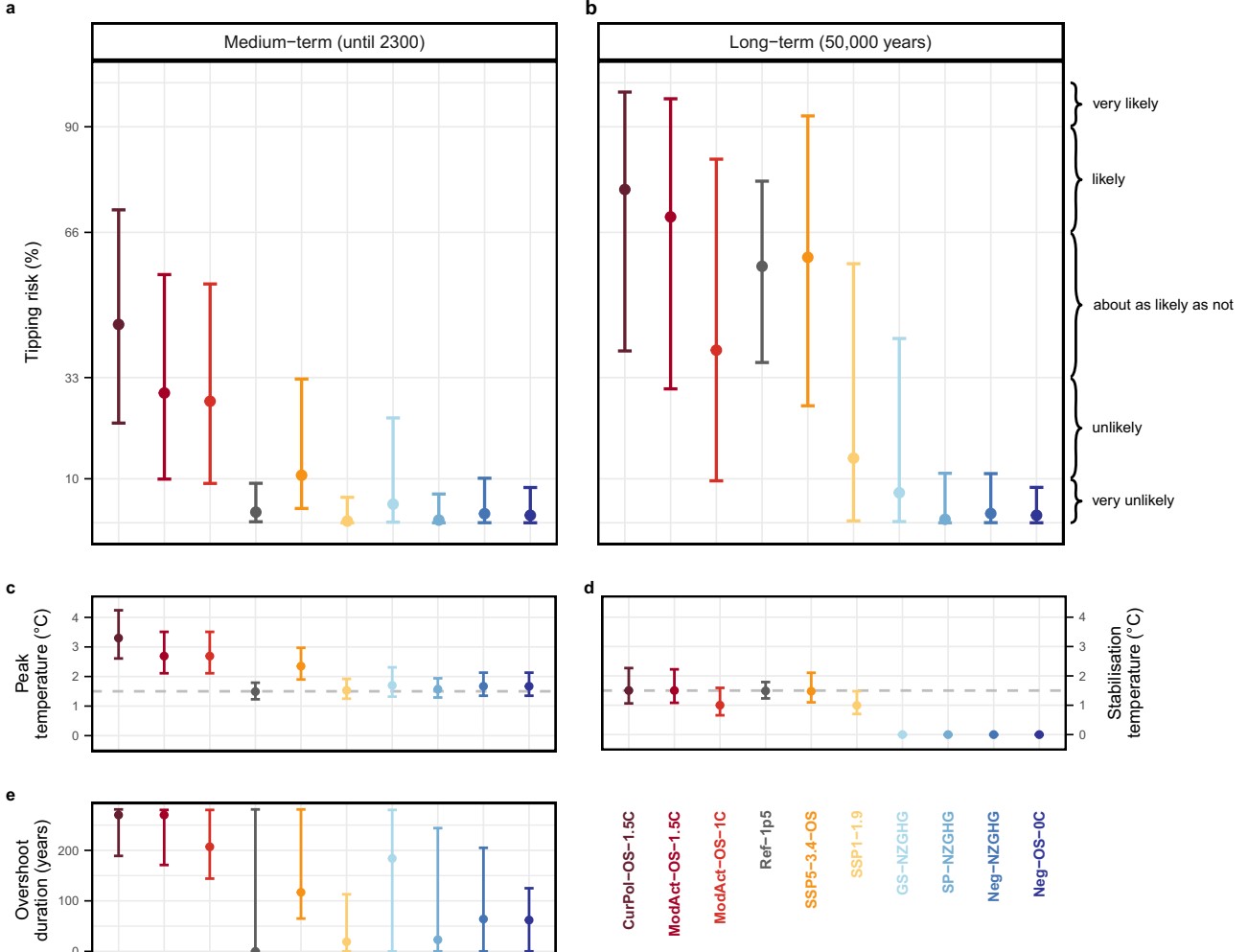

**Fig. 3 | Tipping risk ranges for all emission scenarios. a** In the medium-term (until 2300) and **b** in the long-term (50,000 years), with the risk derived from the median temperature trajectory as centre dots and the range spanning the 10-90th temperature percentiles. IPCC likelihood ranges are given on the right[72]. **c** Peak temperature of the overshoot, **d** long-term stabilisation temperature relative to pre-industrial, with 1.5 °C as a dashed line, and **e** duration of the overshoot above 1.5 °C until 2300.

The contributions of the individual tipping elements to overall tipping risk increase are resolved in Fig. 5b. We find that while AMOC is the main driver of tipping risk increase at lower mean peak temperatures, the AMAZ is the main driver of the non-linear acceleration in tipping risk above 2.0 °C mean peak temperature. This can be explained by the onset of the AMAZ tipping threshold range at 2.0 °C (see Supplementary Table 1). However, the non-linear acceleration at ~2.0 °C mean peak temperature is also observed for the other tipping elements to smaller degrees (see also Supplementary Fig. 6b). As an AMAZ tipping does not drive interactions in our model (compare Fig. 2c), network effects enhancing this behaviour are driven by ice sheet or AMOC tipping (see Supplementary Fig. 6 for the impact of interactions).

The same analysis was conducted with an alternative metric to quantify overshoot, defined by the warming during the overshoot averaged over the overshoot duration (see Supplementary Figs. 7–10). The results are similar to the use of peak temperature.

**Maintaining net zero greenhouse gas emissions to limit long-term tipping risks**

We evaluate the impact of the long-term adherence to achieving and maintaining at least NZGHG emissions on tipping risk for a wide range of climate outcomes per emission pathway (Fig. 6). We find that

pathways that achieve at least NZGHG lead to substantially lower tipping probabilities compared to pathways that do not achieve NZGHG (No-NZGHG), or only do so for some time (No-long-term NZGHG, see Fig. 6). In addition, peak temperature appears to be indicative of tipping risk in the medium term. In the long term, stabilisation temperature, determined by long-term emission behaviour, becomes more decisive (Fig. 3d).

All three classes of pathways display higher tipping risk ranges in the long term than in the medium term. For pathways that only achieve and maintain NZGHG temporarily, the tipping risk range in the medium term is close to the range of the pathways that maintain NZGHG. In the long-term, however, these No-long-term-NZGHG scenarios reach significantly higher tipping risks. For NZGHG temperature trajectories, the median tipping risk remains below 2%, and only for a small number of high-warming trajectories, the risk exceeds 6%.

Our results demonstrate that in order to minimise tipping risks in the long term, it is crucial to achieve at least NZGHG by 2100 as set out in Article 4.1 of the Paris Agreement and maintain it in the long term.

## Discussion

Our study reveals that following current climate policies until 2100 may lead to high tipping risks even if long-term temperatures return to 1.5 °C by 2300. Under such an emission pathway, we report a tipping

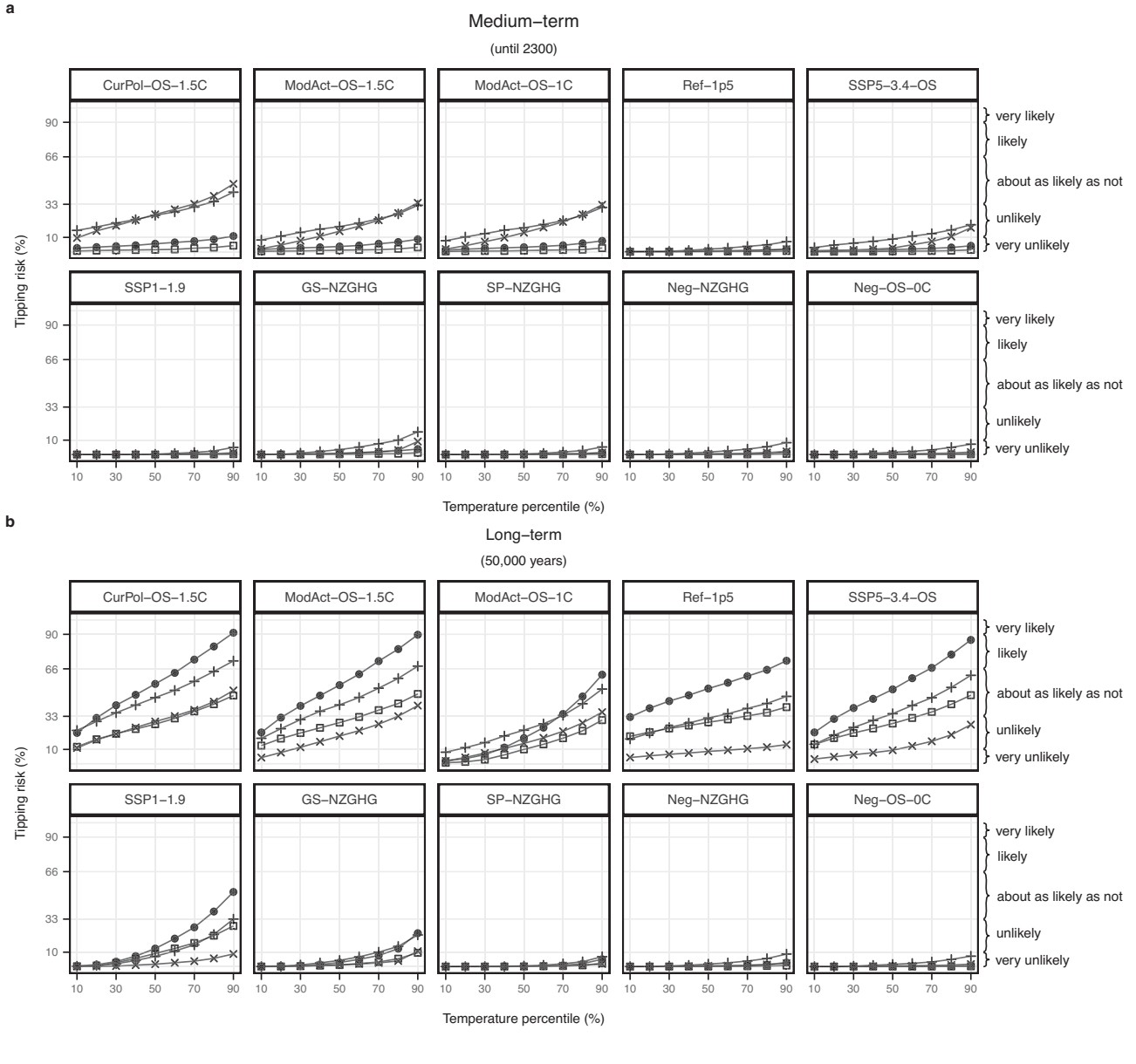

**Fig. 4 | Tipping risk for each of the four investigated core climate tipping elements, Greenland Ice Sheet (GIS), West Antarctic Ice Sheet (WAIS), Atlantic Meridional Overturning Circulation (AMOC), Amazon Rainforest (AMAZ).** **a** Medium-term tipping risk (until 2300). **b** Long-term tipping risk (model equilibrium). The *x*-axis accounts for the uncertainties in climate response, with a 90% probability of the temperature outcome exceeding the lower bound (10th percentile), and a 10% probability of the temperature outcome exceeding the upper bound (90th percentile). The *y*-axis denotes the tipping risk. IPCC likelihood ranges are given on the right[72].

probability of 45% (median estimate, 10–90% range: 23–71%) until 2300 and of 76% (median estimate, 10–90% range: 39–98%) in the long term. Scenarios following pledged NDCs under the UNFCCC in 2020 until 2100 fail to adhere to the Paris Agreement LTTG, and even when subsequently designed such that temperatures return to 1.5 °C (median) after overshoot, we find that they are insufficient to avoid tipping risks (median estimate: 30%, 10–90% range: 10–56% until 2300). We find that tipping risk increases with every 0.1 °C of overshoot peak temperature. Further, we find a non-linear acceleration in tipping risk for peak overshoot temperatures above 2.0 °C resulting in more than 3% tipping risk increase per additional 0.1 °C peak temperature for overshoot temperatures exceeding 2.5 °C peak warming. This underscores the importance of the Paris Agreement climate

objective[24] to hold warming to 'well below 2 °C' even in case of a temporary overshoot above 1.5 °C.

Our results show that only achieving and maintaining net zero greenhouse gas emissions, associated with a long-term decline in global temperatures, effectively limits tipping risks over the coming centuries and beyond in line with earlier studies[2,8,13]. Our findings imply that stabilisation of global temperatures at or around 1.5 °C is insufficient to limit tipping risk in the long term. In order to effectively minimise this risk, our study suggests that temperature needs to return to below 1 °C above pre-industrial level.

There is considerable uncertainty in the response of the climate system to the decline of emissions, and it is not clear how reversible GMT is after emissions cease[41–44]. Regional climate responses show

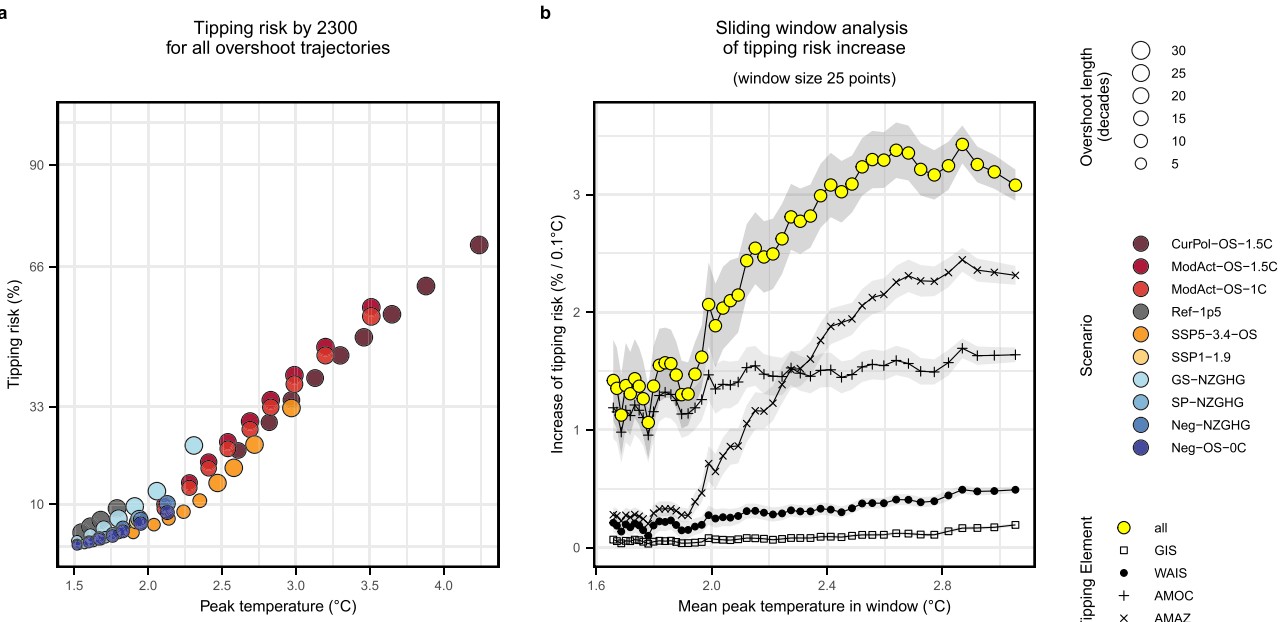

**Fig. 5 | Impact of overshoot peak temperature and non-linear acceleration in tipping risk. a** Increase in tipping risk (%) until 2300 per overshoot peak temperature, for all trajectories with overshoot above 1.5 °C. Each point represents one temperature percentile (10–90%) of a scenario and is coloured by the corresponding scenario information. **b** Acceleration in tipping risk for overshoot peak temperature. Each point represents the slope of a linear fit through a window of 25 adjacent data points of peak temperature vs. tipping risk (see panel **a**), thereby denoting the increase in tipping risk for this window, against the mean peak temperature within this window. The sliding window analysis is shown for all four tipping elements separately: Greenland Ice Sheet (GIS), West Antarctic Ice Sheet (WAIS), Atlantic Meridional Overturning Circulation (AMOC), Amazon Rainforest (AMAZ), as well as for the combined risk of the four considered tipping elements (panel **b**, yellow points). Shaded areas represent the 95% confidence interval.

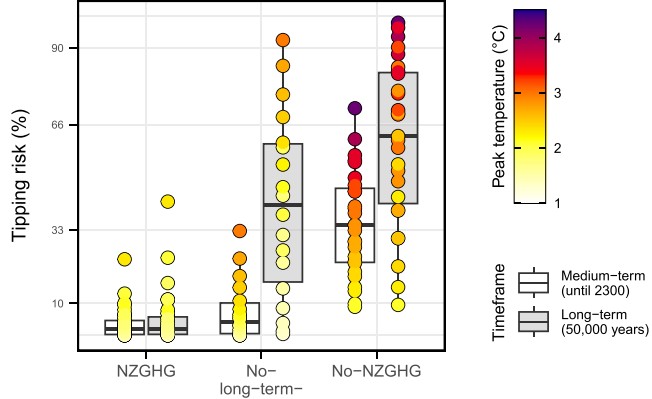

**Fig. 6 | Tipping risk assessed by adherence to the net zero greenhouse gas (NZGHG) criterion.** Each point represents one temperature percentile (10–90%) of a scenario and is coloured by the peak temperature increase. Scenarios were grouped by their adherence to NZGHG ('NZGHG': reach NZGHG emissions by 2100 and maintain NZGHG emissions in the long term; 'No-long-term-NZGHG': reach NZGHG emissions by 2100, but do not maintain NZGHG emissions in the long term; 'No-NZGHG': do not reach NZGHG emissions by 2100) and assessed for both investigated timeframes. Point size is fixed. White boxes indicate the medium-term, grey boxes the long-term, with the upper and lower box edges of the boxplots corresponding to the interquartile ranges of the 25th and 75th percentiles of points per class and the line denoting the median.

high variability indicating that regional climatic changes might only be partially reversible[45,46]. Further, we cannot exclude that reinforcing feedbacks, which will ultimately lead to tipping, have already been triggered in the slow-onset cryospheric tipping elements[4,22]. The transient nature of an overshoot might offer a window of opportunity to counteract anthropogenic emissions with rapid interventions and stabilise the ice sheets before tipping is locked-in[22,47]. Possibilities of recovery and ways to recognise when a transition becomes locked-in and thereby truly irreversible are urgent topics for future research.

While we assess the probabilities of at least one element tipping on the basis of mitigation behaviour until 2300, the implications of overshooting 1.5 °C will unfold over millennia[15]. For example, Global Mean Sea Level will continue to rise for up to 10,000 years or more after emissions have reached NZGHG, due to the slow response of the ice sheets of Greenland and Antarctica[15]. The Global Mean Sea Level Rise (GMSLR) by 2300, committed from historic and currently pledged emissions until 2030, already amounts to 0.8–1.4 m[48]. Exceeding 1.5 °C may lead to a commitment of at least 2–3 m GMSLR on a timescale of 2000 years, and 6–7 m commitment on a 10,000-year timescale[15].

The GMT changes used to assess the tipping risk in this study are derived from emission scenarios with FaIR[35], a simple climate model that is calibrated extensively to match observations and more complex model outputs[49]. Our risk assessment, however, neglects direct temperature feedbacks from destabilising tipping elements, e.g. from disintegrating GIS or WAIS[50] and does not include carbon releases from the AMAZ or permafrost thaw[51–54]. Some of these effects are implicitly accounted for via the uncertainty in the climate response included in this study.

Our stylised Earth system model is designed for risk assessment under large uncertainties on climate tipping elements. As a simplification of the complex climate system, it does not allow us to make exact predictions about the characteristics of tipping[7,13,34]. We do not account for potential multistability, complex path-dependency, or spatial pattern formation[47,55,56]. Furthermore, processes that have the potential to further amplify risks, such as rate-induced tipping as recently suggested for the AMOC[57], are not considered in our study. Anthropogenic influences other than GMT increase, such as changes in land-use[58], are not part of the modelled dynamics, however they enter

implicitly via the assumptions of some of the scenarios used in this study (for instance SSP1 and SSP5[58,59]). These limitations render our results conservative, suggesting that tipping probabilities may well be even higher than we have found. This further underscores the need for a preventive approach to minimise overshoot. The scientific community is working towards more comprehensive and physically based models for the analysis of tipping dynamics, addressing and resolving some of these concerns e.g. under the Tipping Point Modelling Intercomparison Project (TIPMIP)[60]. While this work is under development, we here provide initial results and insights into which scenarios could be interesting to analyse in comprehensive models.

The available quantifications of interactions are taken from an expert elicitation[5] and present a major uncertainty. It would be desirable to constrain this uncertainty better with further analyses, to include more tipping elements, as well as process-based dynamics. However, by including the uncertainties associated with climate sensitivity, carbon-cycle feedbacks, and emissions, and by propagating the uncertainties associated with the tipping elements, our assessment allows for robust results on the tipping risks induced by current mitigation levels and relevant policy scenarios.

All scenarios in this study that fulfil and maintain NZGHG by 2100 rely on carbon dioxide removal (CDR) to varying extents to complement emission reductions to achieve peak warming and allow for a decline in warming thereafter[61,62]. Large-scale deployment of CDR comes with its own concerns[63], depending on the portfolio of CDR technologies deployed. Relying on mitigation technologies that have not yet been deployed at scale is risky[64]. Extensive reliance on land-based CDR options raises sustainability concerns, including competition for land used for food production[65] and impacts on terrestrial and marine biodiversity[66]. Some CDR techniques, such as afforestation will be threatened by climate change itself[67]. Beyond these concerns, deploying CDR at scale will lead to substantial economic costs[65] and unavoidably involve debates on fairness and equity[68].

The lowest need for CDR in our scenario selection is assumed in the SP-NZGHG scenario[69], which contains very stringent reductions in global GHG emissions already by 2030, through a combination of strong policy interventions across multiple dimensions together with ambitious lifestyle changes. Under this scenario, substantial progress along the social and developmental dimensions would be undertaken without further exacerbating environmental degradation. However, substantial gaps in the fulfilment of all dimensions of this scenario remain due to inertia in existing systems and lack of global action[69].

In conclusion, our study shows that current policies and NDCs are not sufficient to minimise tipping risks, even if strong emission reductions after 2100 were to return temperatures to or below 1.5 °C in the long term. Every 0.1 °C of additional overshoot above 1.5 °C increases tipping risk, and greenhouse gas emissions need to reach net zero as early as possible and maintain it to minimise the risk of climate tipping points.

Our results emphasise the fundamental relevance of the Paris Agreement climate objectives[24,62] for planetary stability. To effectively limit tipping risks, holding warming well below 2 °C at all times is essential even in case of a temporary overshoot above 1.5 °C. Beyond peak warming, achieving and maintaining net zero greenhouse gas emissions is paramount to limiting long-term tipping risks by bringing temperatures back down below 1.5 °C and beyond. Our results also illustrate that a global mean temperature increase of 1.5 °C is not "safe" in terms of planetary stability but must be seen as an upper limit. Returning to levels substantially lower, in the long run, might be desirable to limit tipping risks as well as other time-lagged climate impacts such as sea-level rise[15,48]. Domestic policies to reduce emissions need to be adopted and implemented, not only pledged[27], and a more significant and urgent effort is needed to mitigate the risks associated with tipping elements.

## Methods

### Tipping risk and interacting tipping elements
In this study, we classify an element to be tipped once it has transgressed from an untipped to a tipped state at $x > 0$ (see Fig. 1). Further, we define as tipping risk the probability that at least one of the four interacting tipping elements (AMOC, AMAZ, GIS, WAIS) has crossed its tipping point. We obtain this probability through a large-scale Monte Carlo ensemble approach that allows us to account for all parameter uncertainties arising from the tipping thresholds, timescales, interaction strengths, and directions by running the model with a large number (here 11,000) of different parameter combinations (see Fig. 2 for parameter ranges) for every temperature trajectory (evaluating 9 trajectories per emission scenario to account for uncertainties in climate sensitivity) and analysing every ensemble run according to the above-described criteria to assess the states of the four tipping elements at the time of evaluation. The tipping risk is then the percentage of ensemble runs in which at least one tipping element is classified as tipped.

### Scenario classification
We select ten emission pathways from the PROVIDEv1.2 ensemble[32] to span a range of emission reductions (see Fig. 2a). For each of these pathways, we use the resulting probabilistic GMT trajectory (assessed using FaIR v.1.6.2[35]) (see Fig. 2b, Supplementary Fig. 2) to force a model of interacting tipping elements[34] designed to explore different near-term overshoot pathways, peak warmings, and long-term behaviour. We consider the full percentile uncertainty of the PROVIDE scenarios representing equilibrium climate sensitivity of 2.01–4.22 °C (5–95% range) per $CO_2$ doubling[35], resulting in temperature trajectories that may deviate by more than 0.5 °C from the median.

The scenarios were chosen with policy relevance in mind, representing different levels of mitigation and thereby leading to different magnitudes and lengths of overshoot above the LTTG of the Paris Agreement (see Table 1). We classify the scenarios into three groups according to whether they achieve NZGHG emissions by 2100—as set out in the Paris Agreement Article 4.1—or not, and whether they maintain NZGHG in the long term: (i) 'No-NZGHG', (ii) 'No-long-term-NZGHG', and (iii) 'NZGHG' scenarios. NZGHG is understood here as achieving net zero Kyoto GHG emissions, i.e. $CO_2$, $CH_4$, $N_2O$, $SF_6$, HFC, and PFC emissions, as aggregated with the GWP100 metric[36]. We classify the scenarios that reach net zero emissions by 2100, however, beyond this century return to positive emissions that lead to constant rather than declining long-term temperature as 'No-long-term-NZGHG'. The scenarios that do not reach net zero emissions by 2100 are classified as 'No-NZGHG'. In our selection, these scenarios all employ large amounts of negative emissions from about 2130 until temperatures have stabilised at long-term levels at 1.5 or 1 °C, respectively, meaning low positive emissions from 2300 onwards.

### Temperature series extension protocol
The PROVIDEv1.2 time series were linearly extrapolated beyond the year 2300, by either continuing with the stabilisation temperature, if reached in 2300, or otherwise continuing the temperature trajectories with the average slope of the time series per scenario in the period 2290–2300 until they return to 0 °C temperature increase relative to the 1850–1900 average ('preindustrial'), remaining stable thereafter.

### Modelling and propagating uncertainties of coupled climate tipping elements
The dynamics of the four interacting tipping elements (GIS, WAIS, AMOC, AMAZ) are governed by a well-established stylised coupled statistical model[13,34] based on the following set of coupled ordinary

differential equations:

$$\frac{dx_i}{dt} = \left[ -x_i^n + x_i + \frac{n-1}{\sqrt[n]{n^n}} \cdot \frac{\Delta \text{GMT}}{T_{\text{crit,i}}} + d \cdot \sum_{j, j \neq i} \frac{s_{ij}}{10} (x_j + 1) \right] \frac{1}{\tau_i} \quad (1)$$

with $n$ an odd integer; we here use $n = 3$ and perform an additional sensitivity analysis to the exponent presented in the Supplementary materials, using $n = 5, 7$ (Supplementary Fig. 5).

In this model, the state of each of the tipping elements $i$ is denoted by $x_i$. $x_i$ is divided into a baseline state $x_i \simeq -1.0$ and a tipped state $x_i \simeq +1.0$. We define an element to be tipped at time $t$ if $x_i(t) > 0$. The tipping thresholds in terms of global mean temperature increases $\Delta \text{GMT}(t)$ are represented by $T_{\text{crit,i}}$ (see Fig. 2d, Supplementary Table 1). The time-scale parameter $\tau_i$ denotes the tipping timescale that an element needs to transition from its fully functional state to its fully tipped state. The values for $\tau_i$ vary over several orders of magnitude among the four tipping elements (see Fig. 2e, Supplementary Table 1).

The interactions between different pairs of tipping elements are modelled by the last term of Eq. (1). The link strength values $s_{ij}$ are taken from an expert elicitation[5], and each represents a physical mechanism (see Supplementary Table 2). While these link strength values are quantified as relative strengths[5], the absolute importance of the interaction is not known for many of the interactions. Therefore, we introduce the interaction-strength parameter $d$, which is varied between 0 and 1.0, where $d = 0$ means no interaction between the tipping elements and $d = 1.0$ means that the upper limit of any one interaction is of the same order as the strength of the individual dynamic of the tipping element. The prefactor 1/10 sets the coupling term to the same scale as the individual dynamics term by normalising $s_{ij}$ (where $s_{ij}$ is limited to ±10) when $d$ is varied between 0 and 1.0.

Setting the upper boundary of $d = 1.0$ for the maximum interaction strength has the following rationale: If interaction values go beyond 1.0, this will lead to scenarios where the interactions between the tipping elements dominate the state of the climate system, i.e. to cases where the tipping of one element nearly always causes a global cascade of tipping events. Paleoclimate observations indicate that functioning ocean currents and rainforests may be present even in light of disintegrated ice sheets on Greenland and Antarctica[70]. A value of $d > 1.0$ therefore appears implausible. We have included a sensitivity analysis of tipping probability to the parameter $d$ per scenario in our study (see Supplementary Figs. 3, 4), to estimate the relative importance of interactions for tipping probabilities in our approach.

In order to quantify tipping probabilities, we propagate all relevant uncertainties (see Supplementary Tables 1 and 2) in the individual tipping element parameters (tipping thresholds $T_{\text{crit},i}$, tipping timescale $\tau_i$) as well as in their interaction strength described by the parameters $s_{ij}$ and $d$. As the uncertainties are considerable, we need a substantial number of Monte Carlo simulations to capture their effects accurately. The values of the tipping element uncertainties are sampled using a Quasi-Monte Carlo approach based on a latin-hypercube sampling[71]. This reduces the number of required simulations while at the same time, the uncertainty space is covered extensively. Overall, we consider 1000 individual ensemble members that vary in their tipping thresholds $T_{crit,i}$, tipping timescales $\tau_i$, and interaction strength $s_{ij}$. This number is multiplied by 11 for the global coupling strength ($d = 0.0, 0.1, ..., 1.0$). Lastly, all of these 11,000 ensemble members are run through the 10 PROVIDE scenarios, which are all separated into 9 temperature percentile trajectories. This leaves us with 990,000 simulations overall, with an additional 297,000 simulations for the sensitivity analysis to the exponent $n$ of the individual dynamics term (see Supplementary Fig. 5). The analysis presented in the results section is based on the averaged probabilities across the full variation of the global coupling strength $d$. Scenario risk profiles for the full range of outcomes depending on $d$ can be found in the Supplementary (see Supplementary Figs. 3 and 4).

## Data availability
The data necessary to reproduce the findings of this study is freely available (CC-BY-4.0 license) at GitHub via Zenodo at https://doi.org/10.5281/zenodo.8233417. In case of questions or requests, please contact T.M., A.E.H. or N.W.

## Code availability
The code necessary to reproduce the findings of this study is freely available (CC-BY-4.0 license) at GitHub via Zenodo at https://doi.org/10.5281/zenodo.8233417. In case of questions or requests, please contact T.M., A.E.H. or N.W.

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

## Acknowledgements

We want to thank our colleague Gaurav Ganti for his helpful comments and support. The study originated in the master course *Earth System Science & Anthropocene* taught by J.Roc., J.F.D., N.W., and N.H.K. at the University of Potsdam in the summer semester of 2022. T.M., C.F.S., R.D.L. and J.Rog. acknowledge support from the European Union's Horizon 2020 research and innovation programmes under grant agreement No. 101003687 (PROVIDE). N.W., N.H.K., J.Roc. and J.F.D. acknowledge support from the European Research Council Advanced Grant project ERA (Earth Resilience in the Anthropocene, ERC-2016-ADG-743080). N.H.K. is grateful for financial support from the Geo.X Young Academy. J.F.D. is grateful for financial support from the German Federal Ministry for Education and Research (BMBF) in the project 'PIK_Change' (grant 01LS2001A). The authors gratefully acknowledge the European Regional Development Fund (ERDF), the German Federal Ministry of Education and Research and the Land Brandenburg for supporting this project by providing resources on the high-performance computer system at the Potsdam Institute for Climate Impact Research.

## Author contributions

T.M. initiated the study. T.M., A.E.H., C.F.S. and N.W. designed the study. T.M., A.E.H. and N.W. led the writing of the manuscript with input from C.F.S., S.B., N.H.K., R.D.L, J.Rog., J.F.D. and J.Roc. N.W. provided the model code. R.D.L and J.Rog. provided scenario data. T.M., A.E.H., and S.B. implemented the model simulations. T.M. and A.E.H. conducted the analysis. T.M., A.E.H., S.B. and N.H.K. prepared the figures. T.M., A.E.H., C.F.S., S.B., N.H.K., R.D.L, J.Rog., J.F.D., J.Roc. and N.W. gave final approval for publication and agreed to be held accountable for the work performed therein. N.W. led the supervision of the study.

## Funding

## Competing interests

The authors declare no competing interests.
