## [Peer Review File · Nature Communications]

Achieving net zero greenhouse gas emissions critical to limit climate tipping risksEditorial Note: This manuscript has been previously reviewed at another journal that is not operating a transparent peer review scheme. This document only contains reviewer comments and rebuttal letters for versions considered at *Nature Communications*. Mentions of the other journal have been redacted.

REVIEWER COMMENTS

Reviewer #1 (Remarks to the Author):

Review of “Achieving net zero greenhouse gas emissions critical to limit climate tipping risks” by Tessa Möller et al. (submitted to Nature Communications)

This paper is a revised version of a manuscript originally submitted to [REDACTED], which I reviewed at the time.

The paper uses an idealised model to assess the impact of a range of emissions pathways, including overshoot and net zero cases, on the probability of crossing four climate tipping points. The model has two components: the FAIR energy balance model used to assess the response of global temperature to various emissions scenarios, and a network model used to assess the four interacting tipping elements, conditional on global temperature. The manuscript presents probabilities of tipping in the mid-term (2300) and long term, conditional on a range of emissions scenarios examined by IPCC. It concludes that the probability of at least one element tipping increases steeply beyond a certain level of global temperature overshoot.

The subject is highly topical. As the world works towards future emissions pathways that reduce climate change risks, the possibility of crossing climate tipping points is one of the important elements of risk that needs to be understood. The approach taken is interesting and as far as I know hasn't been used before, and it produces a potentially important result about the nonlinearity of the tipping response to different overshoot levels.

In this revised version, the authors have responded carefully to my previous review comments, which were mainly of an overarching nature in terms of what can reliably be deduced from the results and how that was presented. I think the new manuscript is substantially improved. However I think there are still some important issues that need to be resolved before the paper can be published, which I have described below. Because of the overall improvement I have included some more detailed comments this time. Overall I think this work should be published, but I recommend further major revision of the manuscript before acceptance.

Overall points:

For me the most interesting result remains the apparent increase in the slope of the tipping probability vs TA-OS curve (Fig. 4), beyond around TA-OS = 1.8C. I think the simplification of Fig 4, and the addition of Supp Fig. 8 make this more convincing (actually Supp Fig. 8 is more convincing than Fig 4 for me). The result is contingent on the choice of mitigation scenarios (PROVIDE) used. While the choice of scenarios seems rational to me, I'm left wondering how robust the result might be. For example, are we just comparing the three top scenarios (Fig. 4b with the rest (Fig. 4a)? Would the result still apply for a scenario that had a large TA-OS due to a large overshoot for a very

short time (e.g. as I think is being discussed in L 307-309)? And why does the curve in Supp Fig 8 start decreasing above TA-OS=2.15C? Some level of understanding (in terms of the model used) is important because this result is (rightly) brought out in the abstract, it's potentially policy-relevant and the reader needs to understand how robustly/widely applicable it is (or at least the limits within which its applicability is demonstrated).

I think it is very important that the manuscript is precise in its definition of what it means by tipping risk or probability. The topic of the paper is one that attracts a lot of media attention, and I expect this paper will do that when eventually published. The use of the term 'tipping' is vivid and engages the media/public, but it is imprecise and the literature is beset by different uses of the term. As I pointed out in my previous review, this leads to many apparent contradictions when individual scientific studies come up with very different numbers for the 'probability of tipping' of a certain element (e.g. for AMOC in the present study vs AMOC in the recent Ditlevsen & Ditlevsen paper). That causes understandable confusion in the minds of the general public, with the potential result that many people may (wrongly) conclude that the science community really doesn't know anything about this topic. To some extent such confusion is inevitable, but I think as scientists working in this area we need to be making really clear and precise statements about the results that are obtained in each individual paper. Then, individual scientists (who are often called on to comment on their colleagues' published work) can quickly understand precisely what an individual paper is saying, and put it in context with other work.

For this paper, I think what is being assessed is a probability that at least one of four candidate tipping elements crosses a value that is more than half way from its baseline state to an alternative stable state, at least for a short period during the term of interest (up to 2300 or up to 50 kyr in the future). But even having read the methods section carefully I'm not 100% certain that my interpretation is right, e.g. is there an element of commitment/resilience time in the definition of the event that's being considered, that I've missed? This confused me, e.g. around L 193. I do think it would make the (scientist) reader's job much easier if such a precise statement could be included prominently in the main text of the paper, rather than having to reconstruct it from the detail. Such a statement would fit well at the end of the Introduction, alongside the description of the mitigation pathways that are being considered.

Sorry to labour this point, but I feel that it's important for a study like this, which is making some bold claims, to be really precise.

The references appear to have become corrupted somewhere in the revision process. This made it difficult to assess some of the text because I couldn't work out what literature was being cited in support of various statements.

Detailed comments:

L35-35 I think "following current policies" needs to be more precisely defined here. The results are important and potentially policy-relevant. Additionally, I think the "current policies" referred to only go up to 2100 and this needs to be clear (all scenarios are extrapolations beyond that point). The same applies in the Discussion (L 289-290).

L44 “faster than the inherent equilibrium timescales of the respective systems”. Is that right? Or is it faster than the timescale of the GMT change?

Several of the figure references in the text haven't been revised to reflect the new figures (e.g. L153, 274, 276). I recommend a review of all the figure references.

L 222 “Risks *from* overshooting 1.5C”?

L 298-300. For me “show” is too definitive here, as it doesn't take account of the caveats (which are more clearly presented now). Again this is about being clear and accurate about what has been shown, without hiding the results beneath a large pile of caveats.

Final paragraph. To me this goes into policy territory, beyond areas that are directly informed by the science in this paper. I personally feel rather uncomfortable about this in a scientific paper (as opposed to an opinion piece). But that is ultimately an editorial decision.

Reviewer #2 (Remarks to the Author):

The authors have responded to my comments in a satisfactory way.

Reviewer #3 (Remarks to the Author):

Möller et al.

This paper presents a conceptual model for the simulation of tipping elements within the Earth system, specifically focusing on the two ice sheets, the Amazon rainforest, and the AMOC. Using an established 'stylised Earth system model', the manuscript asserts that the "risks of tipping by 2300 increase substantially for scenarios with a peak warming above 34 1.8 °C, or prolonged temperature overshoot beyond 2100". These findings are important and timely.

I reviewed this paper previously, for another journal, and whilst I was generally supportive I had two main reservations about the study in its prior form. Having read the revision I am happy that the authors have addressed those concerns adequately. Supp. Info Fig. 1 is a very useful addition in that regard, and goes a long way towards reassuring the reader that the simplified model can generate reliable behaviours.

To that end, I have no further comments.

REVIEWER COMMENTS

Reviewer #1 (Remarks to the Author):

Review of “Achieving net zero greenhouse gas emissions critical to limit climate tipping risks” by Tessa Möller et al. (submitted to Nature Communications)

This paper is a revised version of a manuscript originally submitted to [REDACTED], which I reviewed at the time.

The paper uses an idealised model to assess the impact of a range of emissions pathways, including overshoot and net zero cases, on the probability of crossing four climate tipping points. The model has two components: the FAIR energy balance model used to assess the response of global temperature to various emissions scenarios, and a network model used to assess the four interacting tipping elements, conditional on global temperature. The manuscript presents probabilities of tipping in the mid-term (2300) and long term, conditional on a range of emissions scenarios examined by IPCC. It concludes that the probability of at least one element tipping increases steeply beyond a certain level of global temperature overshoot.

The subject is highly topical. As the world works towards future emissions pathways that reduce climate change risks, the possibility of crossing climate tipping points is one of the important elements of risk that needs to be understood. The approach taken is interesting and as far as I know hasn't been used before, and it produces a potentially important result about the nonlinearity of the tipping response to different overshoot levels.

In this revised version, the authors have responded carefully to my previous review comments, which were mainly of an overarching nature in terms of what can reliably be deduced from the results and how that was presented. I think the new manuscript is substantially improved. However I think there are still some important issues that need to be resolved before the paper can be published, which I have described below. Because of the overall improvement I have included some more detailed comments this time. Overall I think this work should be published, but I recommend further major revision of the manuscript before acceptance.

Thank you for your insightful comments. We are happy that you find our new manuscript is improved and we believe that we fully address your remaining concerns in this next round of revisions.

Overall points:

For me the most interesting result remains the apparent increase in the slope of the tipping probability vs TA-OS curve (Fig. 4), beyond around TA-OS = 1.8C. I think the simplification of Fig 4, and the addition of Supp Fig. 8 make this more convincing (actually Supp Fig. 8 is more convincing than Fig 4 for me). The result is contingent on the choice of mitigation scenarios (PROVIDE) used. While the choice of scenarios seems rational to me, I'm left wondering how robust the result might be. For example, are we just comparing the three top scenarios (Fig. 4b with the rest (Fig. 4a)? Would the result still apply for a scenario that had a large TA-OS due to a large overshoot for a very short time (e.g. as I think is being discussed in L 307-309)?

And why does the curve in Supp Fig 8 start decreasing above TA-OS=2.15C? Some level of understanding (in terms of the model used) is important because this result is (rightly) brought out in the abstract, it's potentially policy-relevant and the reader needs to understand how robustly/widely applicable it is (or at least the limits within which its applicability is demonstrated).

Thank you for raising these important questions.

We agree with your comments about the importance of the acceleration in tipping risk at 1.8 °C mean TA-OS (which corresponds to around 2.0 °C mean peak warming, see inset Supplementary Fig. 10a), and have therefore chosen to include this result in the main manuscript and restructured Fig. 4 accordingly. We have further come to the conclusion that for the purpose of our analysis TA-OS is not necessarily a superior metric, given that it leads to results consistent with overshoot peak temperature (which is encouraging in regards to robustness), while peak temperature is the easier metric to understand and communicate. Therefore, we decided to focus on peak temperature in the main manuscript (see revised Fig. 4) and provide our additional analysis using the TA-OS metric in the supplement (see Supplementary Fig.s 7-10 and lines 760-770). We are convinced that our results are now easier to understand.

Role of scenario selection

Through the consideration of 9 temperature quantiles per emission scenario, our scenario selection covers the space within 1.5 - 4.2 °C peak temperature well, with a variety of pathways in terms of steepness and duration of overshoot. This includes optimistic scenarios, such as SP-NZGHG, slow-response scenarios such as GS-NZGHG, as well as high emission scenarios. The most extreme scenario - designed specifically to test the response of models to extreme changes in emissions (Lamboll et al. 2022) - is the SSP5-3.4-OS scenario (see Fig. 1a, Supplementary Fig. 2).

We have conducted a series of robustness checks and additional analysis to assess the impact of the scenario selection. For this, we iteratively excluded each emission scenario from the sliding window analysis (revised Fig. 4b) once, and found the nonlinearity at 2.0°C overshoot peak temperature to be robust against these exclusions, making sure it is not an artefact that depends on any one scenario alone. The result is also robust against different window sizes. Each 25 point window contains trajectories from at least 5 different scenarios. The diversity is arguably higher in the centre (which is where the non-linear behaviour is strongest), with some windows containing trajectories from up to 8 different scenarios. In the revised Figure 4a, each dot is now coloured by scenario and thereby makes the scenario-affiliation of the trajectories visible.

Contributions of individual tipping elements

We decomposed the tipping risk into the contributions by each individual tipping element and found that the non-linearity is likely more strongly contingent on the choice of tipping elements / tipping parameters than on the scenario selection.

The overall contribution to tipping risk of the ice sheets by 2300 is comparatively low (see Fig. 3) due to their slow timescales, and so this equally holds for their contribution to the increase in tipping risk. The absolute tipping probability of AMAZ is in the same range as for AMOC (see Fig. 3), although AMOC's inherent threshold range starts at 1.4 °C. Thus, AMOC is the

main contributor to tipping risk increase below 2 °C mean peak temperature in the sliding windows, while the steep increase in tipping risk from ~2 °C mean peak temperature is mainly attributable to AMAZ tipping (see revised Fig. 4). This makes sense, as the internal AMAZ tipping threshold ranges from 2-6 °C, and as one of the faster tipping elements (tipping timescale 50-200 years), with all peaking in the temperature trajectories happening before 2150, there is time to respond for many of the ensemble members by 2300 if the threshold is crossed for long enough. However, the non-linearity at ~2 °C mean peak temperature is also observed for the other tipping elements (see Supplementary Fig. 6b) to smaller degrees.

Role of interactions

To understand the importance of interactions for the observed behaviour, we undertook the same analysis with the global interaction strength set to 0 and set to its highest simulated value (0.9) (see Supplementary Fig. 6). We find that overall, the increase is higher for high coupling, but coupling does not cause drastic qualitative changes in behaviour. Only the GIS displays a smaller increase in tipping risk when coupling strength is high, due to the strong stabilising feedback from AMOC. Also, AMOC tipping risk increase above 2 °C mean peak temperature is intensified with higher coupling, likely because of destabilising interactions from the ice sheets. As AMAZ does not drive any interactions (see Supplementary Table 2), the non-linearity in AMOC above 2 °C must be independent from the AMAZ contribution, and AMAZ itself is only coupled to the AMOC (with an unclear link).

Levelling off of the slope above 2.5 °C mean peak temperature

Above around 2.5 °C mean peak temperature in the sliding window analysis, we see a levelling off of the increase in tipping risk in the range 3-3.5 % per 0.1 °C additional warming (see revised Fig. 4b). This may be due to saturation effects, given that the onset of the AMAZ tipping threshold range plays a decisive role for the non-linearity, and due to its fast tipping timescale most of the committed tipping from overshoot will be detectable (i.e. have crossed the threshold in the model) by 2300 - this interpretation is underscored by the comparatively small differences in absolute numbers for AMAZ tipping between medium- and long-term (see Fig. 3). This could explain the more linear increase for higher levels of warming.

The corresponding decrease in the increase of tipping risk apparent above 2.1 °C mean TA-OS (see Supplementary Fig. 9b) is possibly an effect of the TA-OS metric when applied to high overshoot trajectories with post-overshoot stabilisation temperatures above 1.5 °C (which stretches the overshoot time to 2300 as no return below 1.5 °C takes place).

I think it is very important that the manuscript is precise in its definition of what it means by tipping risk or probability. The topic of the paper is one that attracts a lot of media attention, and I expect this paper will do that when eventually published. The use of the term 'tipping' is vivid and engages the media/public, but it is imprecise and the literature is beset by different uses of the term. As I pointed out in my previous review, this leads to many apparent contradictions when individual scientific studies come up with very different numbers for the 'probability of tipping' of a certain element (e.g. for AMOC in the present study vs AMOC in the recent Ditlevsen & Ditlevsen paper). That causes understandable confusion in the minds of the general public, with the potential result that many people may (wrongly) conclude that the science community really doesn't know anything about this topic. To some extent such confusion is inevitable, but I think as scientists working in this area we need to be making

really clear and precise statements about the results that are obtained in each individual paper. Then, individual scientists (who are often called on to comment on their colleagues' published work) can quickly understand precisely what an individual paper is saying, and put it in context with other work.

For this paper, I think what is being assessed is a probability that at least one of four candidate tipping elements crosses a value that is more than half way from its baseline state to an alternative stable state, at least for a short period during the term of interest (up to 2300 or up to 50 kyr in the future). But even having read the methods section carefully I'm not 100% certain that my interpretation is right, e.g. is there an element of commitment/resilience time in the definition of the event that's being considered, that I've missed? This confused me, e.g. around L 193. I do think it would make the (scientist) reader's job much easier if such a precise statement could be included prominently in the main text of the paper, rather than having to reconstruct it from the detail. Such a statement would fit well at the end of the Introduction, alongside the description of the mitigation pathways that are being considered.

Sorry to labour this point, but I feel that it's important for a study like this, which is making some bold claims, to be really precise.

Definition of tipping risk

Thank you for raising this important issue. We agree with your suggestion and have therefore included a Box ("**Tipping risk and interacting tipping elements**") at the beginning of the manuscript. The box outlines how we define tipping and calculate tipping risk in this paper (the probability that / ratio of ensemble members in which at least one of the four analysed tipping elements (AMOC, AMAZ, GIS, WAIS) has transgressed from an untipped to a tipped state). We additionally included two illustrative figures for clarification. We agree that these additions were necessary to enhance the clarity and precision of our work.

Relation to results from other studies

We also appreciate your concerns about the discrepancy in probability of tipping between our results and the Ditlevsen & Ditlevsen 2023 study, which has received a lot of public attention. Although there is a discussion around Ditlevsen & Ditlevsen's methodology for predicting tipping times (see Ben-Yami et al. preprint, Van Westen et al. 2024), our findings do not contradict their prediction. Our large scale Monte Carlo ensemble includes ensemble members where AMOC tipping occurs within this century, which would be consistent with the Ditlevsen & Ditlevsen paper. However, due to the current state of the literature, we believe a more careful approach is appropriate, that takes the uncertainties in tipping times, thresholds, interaction strengths, into account. For this reason, we have chosen a probabilistic approach (see Monte Carlo approach; described in the Methods) that covers many different possible outcomes, e.g. also AMOC tipping scenarios on much longer timescales. We are confident that the inclusion of Box 1 laying out our methodology of defining tipping risk in this study helps to highlight the probabilistic nature of our study.

The references appear to have become corrupted somewhere in the revision process. This made it difficult to assess some of the text because I couldn't work out what literature was being cited in support of various statements.

Many apologies for the botched citations. We have tried to make sure this issue is corrected in the revised version.

Detailed comments:

L35-35 I think “following current policies” needs to be more precisely defined here. The results are important and potentially policy-relevant. Additionally, I think the “current policies” referred to only go up to 2100 and this needs to be clear (all scenarios are extrapolations beyond that point). The same applies in the Discussion (L 289-290).

Thank you for this observation, we have clarified this in the manuscript and now refer to “a scenario that follows current policies until 2100 and returns long-term temperatures to 1.5°C until 2300” (see lines 33-34, 292-298).

L44 “faster than the inherent equilibrium timescales of the respective systems”. Is that right? Or is it faster than the timescale of the GMT change?

Thank you for raising this question. This formulation might indeed be unhelpful. We have adjusted this sentence to reflect the large effect a small change in forcing can have, which is a more accurate representation of tipping behaviour (lines 45-46).

Several of the figure references in the text haven’t been revised to reflect the new figures (e.g. L153, 274, 276). I recommend a review of all the figure references.

Thank you for pointing this out. We have reviewed all figure references to make sure this issue is corrected in the revised version.

L 222 “Risks *from* overshooting 1.5C”?

Thank you for this suggestion. We agree that it is an improvement and have changed the section heading to **Tipping risk by 2300 from overshooting 1.5 °C**.

L 298-300. For me “show” is too definitive here, as it doesn’t take account of the caveats (which are more clearly presented now). Again this is about being clear and accurate about what has been shown, without hiding the results beneath a large pile of caveats.

Thank you for pointing this out. We have focused this paragraph more strongly on our results. By highlighting the non-linear increase in tipping risk for peak temperatures above 2.0 °C and linking this back to “the importance of the Paris Agreement climate objective to hold warming to “well below 2 °C” even in case of a temporary overshoot above 1.5 °C”, we believe our interpretation is now solidly underlined by the evidence presented through our analysis (lines 300-305).

Final paragraph. To me this goes into policy territory, beyond areas that are directly informed by the science in this paper. I personally feel rather uncomfortable about this in a scientific paper (as opposed to an opinion piece). But that is ultimately an editorial decision.

Thank you for insisting on this point. We have focused the final paragraph more strongly on the tipping risk under overshoot scenarios, that we have quantified in our analysis, and once more underscored how our findings relate to the Paris Agreement. We believe that we are

now accurately highlighting our results, while also showing their policy implications (lines 392-402).

Reviewer #2 (Remarks to the Author):

The authors have responded to my comments in a satisfactory way.

We are happy to hear that the reviewer is content with our revisions.

Reviewer #3 (Remarks to the Author):

Möller et al.

This paper presents a conceptual model for the simulation of tipping elements within the Earth system, specifically focusing on the two ice sheets, the Amazon rainforest, and the AMOC. Using an established 'stylised Earth system model', the manuscript asserts that the "risks of tipping by 2300 increase substantially for scenarios with a peak warming above 1.8 °C, or prolonged temperature overshoot beyond 2100". These findings are important and timely.

I reviewed this paper previously, for another journal, and whilst I was generally supportive I had two main reservations about the study in its prior form. Having read the revision I am happy that the authors have addressed those concerns adequately. Supp. Info Fig. 1 is a very useful addition in that regard, and goes a long way towards reassuring the reader that the simplified model can generate reliable behaviours.

To that end, I have no further comments.

We are happy to hear that the reviewer has found their prior concerns adequately addressed by our revisions.

REVIEWERS' COMMENTS

Reviewer #1 (Remarks to the Author):

I thank the authors for their revisions and response to my further comments. I particularly like the new box defining what they mean by tipping in this paper, and the reformulation of Figure 4 in terms of peak warming. I think these changes make the results and method much clearer.

I have no further comments and I am pleased to recommend the paper for publication.